# PARP Inhibitors: A New Horizon for Patients with Prostate Cancer

**DOI:** 10.3390/biomedicines10061416

**Published:** 2022-06-15

**Authors:** Belén Congregado, Inés Rivero, Ignacio Osmán, Carmen Sáez, Rafael Medina López

**Affiliations:** 1Urology and Nephrology Department, University Hospital Virgen del Rocío, 41013 Seville, Spain; ines.rivero.belenchon@gmail.com (I.R.); nachosman79@hotmail.com (I.O.); rantonio.medina.sspa@juntadeandalucia.es (R.M.L.); 2Department of Pathology, Biomedical Institute of Seville (IBIS), 41013 Seville, Spain; csaez1@us.es

**Keywords:** prostate cancer, PARP inhibitors, mutation test

## Abstract

The introduction of PARP inhibitors (PARPi) in prostate cancer is a milestone and provides a pathway to hope in fighting this disease. It is the first time that drugs, based on the concept of synthetic lethality, have been approved for prostate cancer. In addition, it is also the first time that genetic mutation tests have been included in the therapeutic algorithm of this disease, representing a significant step forward for precision and personalized treatment of prostate cancer. The objectives of this review are: (1) understanding the mechanism of action of PARPi in monotherapy and combinations; (2) gaining insights on patient selection for PARPi; (3) exposing the pivotal studies that have allowed its approval, and; (4) offering an overview of the ongoing trials. Nevertheless, many unsolved questions remain, such as the number of patients who could potentially benefit from PARPi, whether to use PARPi in monotherapy or in combination, and when is the best time to use them in advanced or localized disease. To answer these and other questions, many clinical trials are underway. Some of them have recently demonstrated promising results that may favor the introduction of new combinations in metastatic castration-resistant prostate cancer.

## 1. Introduction

Prostate cancer (PCa) is the most common tumor diagnosed in men in the Western World, the third leading cause of cancer death in Europe, and the second in the United States [1,2]. Fortunately, most PCa are diagnosed at their very early stages and can be cured with radical prostatectomy or radiotherapy. However, more than 5% of patients are diagnosed with metastatic disease, and 30–40% of patients will develop biochemical recurrence after treatment intended to cure. A considerable proportion of them will develop metastatic castration-resistant prostate cancer (mCRPC), a condition that leads to death in a short period of time [3]. PCa can be defined as a heterogeneous condition, which ranges from a relatively indolent to an aggressive disease. Primary PCa is characterized by heterogeneity. The disease is often multifocal, and distinct foci can arise as clonally separate lesions with little or no shared driver gene alterations. In contrast, the metastatic foci of PCa seem to arise from a single clone and manifest subclonal homogeneity [4].

Androgen receptor (AR) signaling plays a central role in the disease’s development; therefore, chemical castration, or androgen deprivation therapy (ADT), remains the mainstay of systemic treatment [5]. Several mechanisms try to explain resistance to this therapy, such as androgen receptor (AR) gain-of-function mutations or splice variants (e.g., AR-V7), loss of tumor suppressor genes (e.g., p53, pTEN), and modifications of stromal components into the tumor microenvironment, promoting invasion, neoangiogenesis, and the metastatization process [6].

In recent decades, a large number of novel drugs with different mechanisms of action have been introduced in the therapeutic management against prostate cancer in all stages, focusing mainly on metastatic and castration-resistant disease, with and without metastasis, where they have shown to delay progression and prolong survival. Clinical options include cytostatic agents (docetaxel plus prednisone, cabacitaxel), second-generation antiandrogens, (abiraterone, enzalutamide, apalutamide, and darolutamide) bone-targeted therapies (Radium 223), immunotherapy (sipuleucel-T), Akt inhibitors, and radioisotopes (Lutetium-177–PSMA-617). However, there is a lack of high-quality randomized trials exploring different treatment sequences.

Therefore, there is an urgent need for more effective treatments for patients with mCRPC, especially those who have progressed after novel hormonal therapies and taxane chemotherapies.

In the last two years, two poly (ADP-ribose) polymerase (PARP) inhibitors (Olaparib and Rucaparib) have been approved for mCPRC, opening a new horizon in the treatment of PCa [7]. Olaparib was approved for mCRPC patients with any Homologous Recombination (HR) mutation who had previously received a second-generation hormonal agent, and rucaparib was approved for mCRPC patients with a BRCA1 or BRCA2 mutation who had previously received both a second-generation hormonal agent and a taxane chemotherapy.

The rationale for the use of PARP inhibitors (PARPi) is mainly based on the high frequency of genetic mutations found in PCa and the interesting phenomenon of synthetic lethality. Genetic alterations in PCa include germline and somatic mutations. Germline mutations affect all cells of the body and may be useful for genetic counseling. Somatic alterations occur only in tumor cells and may evolve due to intrinsic genome instability within the tumor and clonal selection resulting from earlier therapies. Germline mutations in Homologous Recombination (HR) DNA repair genes have been observed in approximately 10–15% of patients with metastatic PCa, while somatic mutations occur in 20–25% of those patients, with BRCA2 and ATM being the most frequently mutated. Moreover, compared to locoregional tumors, mCRPC presents a higher frequency of aberrations in the AR, TP53, RB1, and PTEN genes [8]. This evidence supports the research of PARPi in CPRCm. Nevertheless, these genetic alterations have also been described in localized disease, conferring a more aggressive evolution. Therefore, research with PARPi is being extended to earlier stages of the disease.

Synthetic lethality is a genetic concept in which the functional loss of two genes results in cell death, while the functional loss of only one of them is compatible with cell viability (Figure 1).

BRCA1 and BRCA2 are tumor suppressor genes involved in transcriptional regulation and repair of double-strand DNA damage breaks (DSBs) in the DNA molecule, playing a key role on the HR path. Cells with loss-of-function in these genes are unable to repair errors in DNA, depending on the ability of PARP to activate alternative routes. Since they depend on PARPs to maintain genome integrity, these cells are highly vulnerable to PARPi.

There are still many unanswered questions regarding the use of PARPi in PCa, including who would benefit more from a genetic study and when is the best time to use it in the natural evolution of the disease [9]. These questions make it necessary to deepen the knowledge of these drugs to find potential ways to maximize their clinical effectiveness.

This review aims to update the available evidence on PARPi for PCa treatment.

## 2. Rationale of Use of PARP Inhibitors in PCa

### 2.1. DNA Repair Pathway

The integrity of DNA is continuously challenged by a variety of agents and processes that can alter, directly or indirectly, the sequence of this molecule. Damage that arises in DNA, its repair, or its absence are key factors in the appearance of mutations that initiate and promote tumorigenesis [10].

Given the potentially devastating effect of these types of mutations, the healthy cells evolved to defend themselves from the deleterious effects that DNA damage causes through several molecular pathways, which, as a whole, are called DNA Damage Response (DDR). These pathways have the ability to identify the damage, promote the arrest of the cell cycle, and, ultimately, proceed with its repair, contributing to the maintenance of genome integrity.

DDR can be divided into distinct but functionally interconnected means that process the repair of existing damage: first, the repair of double-strand DNA damage breaks (DSB), which includes repair by Homologous Recombination (HR) and nonhomologous recombination; second, the repair of single-strand breaks (SSB), of which the excision repair base (ERB) is a part. Several key proteins contribute to the correct function of these processes. The BRCA1, BRCA2, PALB2, ATM, CHEK1, CHEK2, and RAD51 play a role in the HR process. However, Poly enzymes (ADP-Ribose) Polymerase 1 and 2 (PARP1 and PARP2) are the pillars of the ERB process [11].

Those proteins also play a key role in the repair of DSB by facilitating the activation of the repair by HR and contributing to the inhibition of less conservative repair pathways, such as the nonhomologous route end joining (NHEJ). Thus, the absence of PARPs contributes to the dysfunctional HR process, making nonconservative DNA repair processes the dominant pathways [12].

### 2.2. Role of DNA Damage Repair Genes in Prostate Cancer

#### 2.2.1. DDR Mutations in Prostate Cancer

Classical studies show that germline mutations in HR DNA repair genes have been observed in approximately 10–15% of patients with metastatic prostate cancer, while somatic mutations occur in 20–25% of those patients [13,14].

Of these mutations, BRCA2 mutations are the most frequently found (12–18%), while ATM (3–6%), CHEK2 (2–5%), and BRCA1 (<2%) are other commonly altered genes involved in HRR [15,16].

Since DDR pathway alterations were seen at similar rates between localized and metastatic PCa, it has been speculated that PARPi may also have a therapeutic effect in localized PCa.

To support this hypothesis, a 2019 study by Kim et al. [17] analyzed the DDR pathway alterations in localized PCa using The Cancer Genome Atlas (TCGA) public database. Their results highlighted that the DDR alteration rate was surprisingly higher than suggested by previous studies [18,19] and was associated with shorter OS in men with postoperative HR features.

A recent systematic review of the prevalence of DNA-damage-response gene mutations in prostate cancer that summarizes the prevalence of DDR mutation carriers in the unselected (general) PCa and familiar PCa populations confirms these findings [20].

BRCA1/2 genes are located at chromosomes 17q21 and 13q12, respectively. They are large genes consisting of 100 and 70 kb, respectively. They have an autosomal dominant inheritance pattern with incomplete penetrance.

#### 2.2.2. Germline Mutations

Multiple studies have reported an association between frequent germline deleterious mutations in DDR genes and advanced PCa. Specifically, germline BRCA1/2 mutations are associated with increased risk and more aggressive PCa, higher risk of nodal involvement, and distant metastasis at diagnosis [21].

Germline BRCA2 mutations increase the risk of developing PCa eightfold at the age of 65 years [22]. In the localized disease, germline BRCA1/2 mutations are related with progression in patients undergoing active surveillance and a high rate of recurrence in patients treated with curative intention [23,24]. Similarly, in the metastatic setting, it is associated with a more aggressive evolution [25].

The prevalence of germline mutations varies among countries and ethnic groups.

The International Stand Up to Cancer/Prostate Cancer Foundation team (SU2C-PCF) studied 150 metastatic PCa identifying 8% of germline and 23% of somatic DDR mutations [13]. BRCA2 was the most frequent mutation (13%), followed by ATM (7.3%), MSH2 (2%), and BRCA1, FANCA, MLH1, RAD51B, and RAD51C.

Pritchard et al. studied germline mutations in 692 men with mCRPC with no family history, identifying 84 deleterious mutations in the 20 DNA repair genes studied in 82 men (11.8%), being BRCA2 the most frequent one (5.3%) [14].

Nicolosi et al. analyzed 3607 men with PCa [26], finding germline mutations in 620 of them (17.2%), of which 30.7% were BRCA1/2 mutations; other mutations included ATM, PALB2, CHEK2, and mismatch repair genes PMS2 and MLH1/2/6.

#### 2.2.3. Somatic Mutations

Somatic mutations also stimulate carcinogenesis [27]. As explained before, Robinson et al. found that 23% of mCRPC had somatic mutations in DNA repair pathway genes, with most mutations in BRCA2 and ATM [11]. Different studies have found 12% and 8% of PCa patients carrying a BRCA1/2 or ATM mutation, respectively, and more frequently in mCRPC [28]. Abida et al. observed somatic BRCA2 mutations in tumors before they progressed to metastatic disease. Somatic BRCA2 mutations have been suggested to arise early in tumors from patients who eventually develop metastatic disease, while ATM alterations seem to enrich in CRPC [16].

### 2.3. Mechanism of Action of PARP Inhibitors

The original rationale for using PARPi as a cancer treatment is that PARPi can sensitize tumor cells to therapies that cause DNA damage, such as chemo or radiotherapy. The inhibition of PARP-mediated repair of DNA damage produced by chemotherapy or radiotherapy, may result in an increased of therapeutic potency [29].

Almost two decades ago, two groups described the important concept of Synthetic Lethal (SL) interaction between PARP inhibition and BRCA1 or BRCA2 mutation, which represented a new therapeutic option for BRCA-mutant tumors [30,31].

SL means that a defect in either one of two genes has little effect on the organism, but a combination of defects in both genes results in cell death.

Carriers of deleterious heterozygous germline mutations in the BRCA1 and BRCA2 genes have high risk of different types of cancer, such as PCa [32]. BRCA1 and BRCA2 are tumor-suppressor genes involved in transcriptional regulation and, as stated before, are critical to the repair of DSBs in the DNA molecule, playing a key role in the HR pathway [31]. Cells with functional loss in these genes are unable to repair errors in DNA, depending on PARPs ability to detect these damages and activate alternative repair pathways.

By depending on PARPs to maintain genome integrity, these cells are highly vulnerable to PARPi. Although the functional loss of only one of the genes is tolerated (BRCA or PARP), the inhibition of PARP in cells with BRCA mutations makes them unable to repair DNA damage, causing the accumulation of errors and, ultimately, leading to cell death (synthetic lethality) [11].

PARPi antitumor activity is based on the concept of synthetic lethality, in which two separate molecular pathways, which are not lethal when disrupted individually, cause cell death when inhibited simultaneously (Figure 2).

The demonstration that BRCA-mutant tumor cells were as much as 1000 times more sensitive to PARPi than BRCA-wild type cells provided the impetus for PARPi to be tested in clinical trials as single agents. PARPi were the first clinically approved drugs to explore the SL mechanism [33]. This concept has been reviewed, since it has been shown that some PARPi “trap” PARP1 on DNA, preventing autoPARylation and PARP1 release from the site of damage, thereby interfering with the catalytic cycle of PARP1 [34,35,36].

PARPi differ in their capacity to trap PARP1; talazoparib is nearly 100 times more potent than niraparib, followed by olaparib and rucaparib. However, veliparib has a low ability to trap PARP1. These differences in PARP1 trapping are a predictor of in vitro cytotoxicity in BRCA mutant cells, being talazoparib the one with more cytotoxic capacity [35] (Figure 3).

This different trapping capacity is also important when combined therapies are studied, since a high trapping activity could be accompanied by high toxicity.

Initially, it was postulated that PARPi would only be effective in tumors with mutations in the BRCA1/2 genes. However, in more recent clinical studies, it was found that tumors without BRCA1/2 mutations might respond to this treatment, although its benefits are less significant [37,38].

The “trapping” effect of PARPi can also have a direct cytotoxic effect: PARPi will trap PARP proteins to the ssDNA damage sites, preventing its release, which is essential for mediator proteins to initiate repair. The trapped PARP proteins will cause replication fork stalling, which will increase replication stress and eventually cause cell death [39].

Therefore, based on these observations, the number of patients who could theoretically benefit from PARPi has been widely extended. PARPi can be used as monotherapy or in association with other drugs, namely in combination with chemotherapy, immunotherapy, and other targeted therapies that limit DNA damage repair. The combination of drugs has a synergistic and advantageous effect, to the extent that it makes it possible to overcome the resistance to PARPi and increase the effectiveness of these drugs [40].

Table 1 represents the chemical structures and pharmacological properties of the currently available PARP inhibitors (PARPi).

## 3. Clinical Development of PARP Inhibitors in Prostate Cancer

### 3.1. PARPi in Monotherapy

#### 3.1.1. OLAPARIB

This was the first drug to be developed in this group, initially for breast and ovarian cancer, and subsequently in pancreatic and PCa.

The phase II TOPARP trial provided some of the first real evidence for the development of PARPi in advanced PCa. TOPARP-A [41], evaluated the response of Olaparib among 49 mCRPC men, defining response as radiological response according to Response Evaluation Criteria in Solid Tumours version 1.1 [RECIST 1.1], a PSA response >50%, and/or a confirmed circulating tumor cells (CTC) count conversion from high to low counts. 33% patients were classified as responders. Germline or somatic mutations in DNA repair genes (including BRCA1, BRCA2, ATM, or PALB2) were analyzed among responders.

The TOPARP-B phase II trial selected 98 mCPRC patients who had received taxane-base chemotherapy with genetic sequencing of prostate cancer biopsies showing DDR gene aberrations (31% with the BRCA2 mutation) and prospectively randomized them to receive 300 mg or 400 mg of olaparib twice daily. Responses were obtained in all mutations of the HR spectrum, being clearly superior in BRCA2 (83%) and followed by PALB2, ATM, and the CDK12 mutation, which was the lowest [42].

Only olaparib has been approved based on data from a randomized phase III trial, the PROFOUND study [43]. This trial evaluated olaparib in men with mCRPC with genetic alterations in DDR mechanisms (mainly BRCA1, BRCA2, and ATM) and disease progression after enzalutamide or abiraterone. Patients with mutations were randomized to receive olaparib versus abiraterone or enzalutamide. The trial was structured around two cohorts: cohort A included 245 patients with mutations in BRCA1, BRCA2, or ATM; cohort B included 142 patients with alterations in any of the 12 other prespecified genes in the study protocol (BRIP1, BARD1, CDK12, CHEK1, CHEK2, FANCL, PALB2, PPP2R2A, RAD51B, RAD51C, RAD51D, and RAD54L).

The study was positive with an improvement of the imaging-based progression-free survival (rPFS) in cohort A (primary endpoint), which was longer in the olaparib group (7.4 months vs. 3.6 months, Hazard Ratio (HR) 0.34, 95% CI 0.25–0.47, *p* < 0.001). Objective response ratio (ORR) and overall survival (OS) were also higher in the olaparib group of cohort A: 33% vs. 2%, and 19.1 months vs. 14.7 months, (HR 0.69, 95% CI 0.50–0.97, *p* = 0.0175), respectively, while 67% of the patients in the control arm crossed over to receive olaparib, following radiographic progression.

On the other hand, there was no statistically significant benefit in PFS for the combined cohort of patients harboring any of the 15 prespecified HRR gene mutations.

An unplanned exploratory analysis found that the benefit may be greater for those patients who receive olaparib before taxane-chemotherapy.

In May 2020, olaparib received the United States Food and Drug Administration (FDA) approval for the treatment of mCRPC in progression to antiandrogens (abiraterone and enzalutamide) in patients with a somatic mutation in any HRR gene or any germline mutations in BRCA 1, 2, and ATM genes, and germline and somatic testing for relevant alterations are now considered to be the standard of care for these patients [44]. In Europe, however, EMA approval is restricted to patients with BRCA1/2 alterations.

An unresolved question is whether PARPi are efficacious in hormone-sensitive (noncastrate) PCa in the absence of androgen suppression. To that end, a study tested the use of olaparib in men with biochemically recurrent PCa following prostatectomy without concurrent androgen deprivation therapy. In this trial of genomically unselected patients with biochemical recurrence, interim results from the first 20 patients demonstrated that three (15%) had PSA responses and four (20%) had minor PSA responses. Interestingly, the three patients with PSA responses (of which two had complete responses) had germline or somatic BRCA2 mutations. That study is now being expanded to enroll a total of 50 men with biochemically recurrent PCa [45].

#### 3.1.2. RUCAPARIB

Although rucaparib has not been as well-studied as olaparib, a nonrandomized single-arm phase II study (TRITON2) showed positive results for this drug. In that study, 62 patients with mCRPC with HR mutations, pretreated with antiandrogen and docetaxel, were treated with rucaparib. In the BRCA1 or BRCA 2 mutated subgroup, the RR was 44–50% (95% CI: 31.0–56.7%; 27 of 62 patients), including three complete responses, and the confirmed PSA response rate (RR) was 54.8% (95% CI: 45.2–64.1%; 63 of 115 patients). The median duration of response was not reached at the time of analysis (between 1.7 to 24 months) [46]. In the patients without BRCA mutations, the RR was clearly lower [47,48]. This study led to FDA approval of rucaparib in 2020, conditional on the results of the phase III trial TRITON3. This ongoing trial randomizes patients with mCRPC and BRCA 1/2 or ATM mutation in progression to a hormonal line to receive abiraterone, enzalutamide, or docetaxel, at the investigator’s discretion, vs. Rucaparib [49].

To study if PARPi are efficacious in hormone-sensitive prostate cancer in the absence of androgen suppression, a study (TRIUMPH) is testing rucaparib in men with metastatic hormone-sensitive prostate cancer (mHSPC) with germline HRR gene mutations [50]. Rucaparib is being used in this study in the absence of androgen deprivation therapy or other novel hormonal agents. The Phase II Study ROAR is investigating rucaparib in monotherapy in nonmetastatic, biochemically recurrent prostate cancer after prior definitive local therapy (prostatectomy or radiation therapy) [51].

#### 3.1.3. NIRAPARIB

Niraparib is a PARP1/2 inhibitor with higher trapping potency and cytotoxicity than olaparib [34].

The phase II GALAHAD evaluated niraparib in mCRPC patients with HRR deficiency, progressing after taxane and a new hormonal agent. HRR deficiency was defined as a biallelic alteration in BRCA1/2, ATM, FANCA, PALB2, CHEK2, BRIP1, or HDAC2, assessed by a blood-based circulating tumor DNA (ctDNA) or tissue-based test. One of the interesting points of this study is the use of a liquid biopsy NGS test for patient selection, since in all the other phase II trials of PARPi in mCRPC, a pathogenic mutation was sufficient to call a patient eligible, regardless of evidence of loss-of-heterozygosity or other means of second allele hits. Patients received 300 mg of niraparib daily. The primary endpoint was the objective response rate (ORR).

Preliminary results were reported for a population of 81 patients with a biallelic HRR deficiency (46 BRCA1/2 and 35 non-BRCA). Of the 51 patients with a measurable disease, the ORR in the “BRCA group” was 41% (95% CI 23.5–61.1%) compared to 9% (95% CI 1.1–29.2%) in the “non-BRCA group”; and the CRR was 63% (95% CI 47.6–76.8%) compared to 17% (95% CI 6.6–33.7%), respectively. Median PFS and OS in BRCA were 8.2 months (95% CI 5.2–11.1 months) and 12.6 months (95% CI 9.2–15.7 months), respectively, versus 5.3 months (95% CI 1.9–5.7 months) and 14.0 months (95% CI 5.3–20.1 months) in non-BRCA [52].

Similar to olaparib, niraparib appears to be more effective in BRCA-mutated patients. Efficacy is similar to that of olaparib (ORR of 52.4% and 41% for BRCA groups in TOPARP-B and GALAHAD, respectively) [53].

Currently, niraparib is being studied as neoadjuvant in a phase II trial before surgery in patients with high-risk localized PCa and DNA DDR [54].

#### 3.1.4. TALAZOPARIB

Talazoparib is a potent inhibitor of PARP. In addition to having a high ability to inhibit the activity of catalytic enzymes, it has greater potency in trapping PARP1 to DNA errors [55].

Talazoparib was assessed in an open-label phase II trial (TALAPRO-1) in patients with mCRPC and DDR-HRR alterations (mutations in ATM, ATR, BRCA1, BRCA2, CHEK2, FANCA, MLH1, MRE11A, NBN, PALB2, or RAD51C).

The primary endpoint was the ORR. Out of the 104 patients evaluable, 50% had BRCA2 mutations, whereas alterations in BRCA1, ATM, or PALB2 accounted for 4, 14, and 4% of patients, respectively. After a median follow-up of 16.4 months (11.1–22.1), the radiological RR in the study was 29.8% (95% CI: 21.2–39.6). Similar to in TOPARP and TRITON2 studies, patients with BRCA1/2 mutations had a higher response rate (46% radiological RR, 66% PSA50 RR, 72% CTC conversions RR).

Talazoparib showed lasting antitumor activity in these heavily pretreated patients with mCRPC and DDR-HHR gene alterations [56].

### 3.2. Adverse Events and Tolerability

Despite presenting a very high safety profile, PARPi have demonstrated, in phase III clinical trials, some adverse effects, of which fatigue, gastrointestinal (GI) symptoms, and myelosuppression are the most common [41]. The main adverse reactions of these drugs are usually mild to moderate, can be managed with dose reductions, and do not require discontinuation of treatment. Fatigue is the most frequently observed adverse effect, and it seems to be transversal to all PARPi. GI adverse effects are extremely common and tend to occur in all patients treated with PARPi. Nausea is the most prevalent adverse effect, occurring in 76% of patients treated with olaparib, 75% with rucaparib, 74% with niraparib, and 49% of patients treated with talazoparib, followed by vomiting, diarrhea, constipation, and abdominal pain.

Hematologic toxicity usually appears early after the start of treatment with PARPi and tends to resolve a few months after taking the drugs. Anemia is the main hematological adverse event, occurring in 44% of patients treated with olaparib, 50% with niraparib, 37% with rucaparib, and in 53% of patients treated with talazoparib. Of all the PARPi, niraparib is correlated with the highest hematological toxicity [37,55,57,58].

According to prostate cancer studies, in the PROFOUND trial, the most common adverse events were hematological (anemia, 46%), gastrointestinal (nausea, 41%, loss of appetite, 30%), and fatigue or asthenia (41%) [43]. The GALAHAD and TRITON2 studies also found that anemia is the most frequent adverse event, 17.9–25% [52,59].

However, combining PARPi with other drugs may increase the likelihood of toxicity. A meta-analysis of 14 trials found that myelodysplastic syndrome can appear in combined therapy, but the incidence is low [60]. Additionally, elevated levels of transaminases have been reported, and creatinine increases were seen in 10 to 15% of patients treated with olaparib, rucaparib, and niraparib, but not with talazoparib. None dose reduction was required.

Olaparib and rucaparib are metabolized by CYP450, so drug interactions are more frequent than with talazoparib and niraparib. Regarding the number of tablets taken per day, it varies from one for talazoparib, two tablets twice a day for rucaparib and olaparib, and three once a day for niraparib. Nevertheless, all these data should be considered with caution, as we have no direct comparisons between different PARPi yet.

### 3.3. Mechanisms of Intrinsic and Acquired Resistance to PARP Inhibitors

Similar to other targeted therapies, resistance to PARPi has been observed in most patients with advanced tumors [61]. There are several mechanisms for resistance proposed so far that demonstrate how tumor cells stop responding to the cytotoxic effects of PARPi, and can be grouped as follows:Resistance mechanisms that restore the pathway of PARP-independent homologous recombination. There are multiple pathways leading to the restoration of HR function, including:
Minor mutations of the genes involved in HR (BRCA1/2, PALB2, RAD51C/D) that restore the ability to repair errors in DNA. This mechanism is also responsible for platinum resistance [62];Expression of hypomorphic variants BRCA1/2;Epigenetic changes in the genes that take part in HR, such as the promoter demethylation of BRCA1 and RAD51C genes [63]; andMutations that compromise the regulation of the DNA end-resection through the loss of 53BP1, REV7/MAD2L2, or Shieldin complex, allowing for the maintenance of HR in the absence of BRCA1 [64,65].Resistance mechanisms independent of the HR pathway:
Stabilization of the replication fork, a process in which the BRCA1/2 and PARP1 proteins have a crucial role. In its absence, the replication fork is not stabilized, no repair of errors in the DNA occurs, consequently leading to cell death [66,67];Reduced expression of PARP enzymes, which contributes to the reduction of drug activity and reduction of PARP1 entrapment [64];Drug efflux due to the expression of ATP-binding cassette (ABC) transporters in the tumor cells, such as P-glycoprotein, increasing the efflux of PARPi, reducing the availability of the drug, and, consequently, reducing its effects [63,65];Patients with reverse mutations in the BRCA1 gene exhibit MMEJ signals, suggesting the POLQ as a PARPi resistance vehicle. Thus, POLQ inhibitors can suppress resistance acquired from PARPi, conferring synthetic lethality in tumors with deficits in HR and NHEJ [67]; andMutations in the poly (ADP-ribose)-glycohydrolase enzymes (PARG) can lead to resistance through a mechanism that does not restore HR. The loss of PARG results in the accumulation of chains PAR which, by not being degraded, maintain the activity of the enzymes.

Since restoration of HR appears to be one of the main causes of resistance, most research has focused on this area. A better understanding and monitoring of such resistances is necessary to design studies combining PARPi with other treatments.

### 3.4. PARPi in Combination

The objectives of combined treatments with PARPi are mainly two: to delay resistance to treatment with PARPi, and to expand the therapeutic target to patients who may be resistant to PARPi in monotherapy [40].

Some combinations are currently being studied.

#### 3.4.1. Combinations with Antiandrogen Therapy

The androgen receptor axis is the main therapeutic target in PCa [68,69]. The study of the combination of the AR and DDR pathways is based on preclinical data that have demonstrated synergism between these two groups of drugs in three key ways:PARP promotes AR transcription, so inhibition of this pathway potentiates the antiandrogenic effect [70];ADT promotes PARP overexpression, increasing its sensitivity to PARP inhibitors; andAntiandrogen therapy also inhibits the expression of genes of the DDR system, producing genomic instability and thus promoting mutations in DDR, generating sensitivity to PARPi. This is known as the BRCA-ness phenotype [71,72,73].

Several combinations of PARPi and antiandrogens have been studied. A randomized phase II trial analyzed the combination veliparib-abiraterone, not finfing differences in outcomes [74].

Other trials have shown positive results. The combination olaparib–abiraterone was assessed in a phase II, double-blind, placebo-controlled study of olaparib plus abiraterone versus abiraterone in patients with mCRPC in progression to Docetaxel. In total, 142 patients unselected for HRR mutations were included. rPFS was significantly increased in the combination arm (13.8 vs. 8.2 mo; HR 0.65, 95% CI 0.44–0.97; *p* = 0.034). This was the first study to report a statistically significant increase in rPFS following combination treatment with a PARPi and androgen receptor-directed treatment in patients with mCRPC [75]. The retrospective analysis of genomic profiles showed that patients with and without HRR mutations benefit from the combination [76].

Kim N Chi et al. have recently presented at the 2022 annual American Society of Clinical Oncology-Genitourinary Cancer Symposium (ASCO-GU) the first results of the MAGNITUDE phase III trial assessing niraparib with abiraterone acetate and prednisone (AAP) as first-line therapy in patients with mCRPC with and without HRR gene alterations. MAGNITUDE (NCT03748641) is a randomized, double-blind phase III study in mCRPC. Patients with HRR biomarker positive (ATM, BRCA1, BRCA2, BRIP1, CDK12, CHEK2, FANCA, HDAC2, and PALB2) and without specified gene alterations (HRR biomarker negative) were randomized 1:1 to receive niraparib 200 mg once daily and AAP or placebo and AAP. The primary endpoint was rPFS assessed by a blinded, independent central review in the BRCA1/2 group followed by all HRR biomarker positive patients. Secondary endpoints were time to initiation of cytotoxic chemotherapy, time to symptomatic progression, and OS. Other endpoints included time to PSA progression and ORR.

With a median follow-up for this analysis of 18.6 months, niraparib and AAP significantly improved rPFS in the BRCA1/2 subgroup, reducing the risk of progression or death by 47% (16.6 vs. 10.9 months) and in all HRR biomarker positive patients by 26% (16.5 vs. 13.7 months; HR 0.74, 95% CI 0.57–0.97). Niraparib and AAP delayed the time to initiation of cytotoxic chemotherapy (HR 0.59, 95% CI 0.39–0.89), time to symptomatic progression (HR 0.69, 95% CI 0.47–0.99), and time to PSA progression (HR 0.57, 95% CI 0.43–0.76), as well as improved ORR in HRR biomarker positive patients. At the time of the first interim analysis, OS data were premature (27% of deaths observed: HR 0.767, 95% CI 0.525–1.119). No benefit was observed with niraparib and AAP in patients who were HRR biomarker negative [77].

Some other exciting results have been presented at the 2022 ASCO-GU. Saad F. presented results of the phase III double-blind PROpel trial, in which 796 men with ongoing androgen deprivation therapy for mCRPC were randomly assigned to receive first-line treatment with either olaparib 300 mg twice a day or matching placebo alongside AAP or prednisolone at a dose of 5 mg twice daily. The planned interim analysis showed that the addition of olaparib to abiraterone improved the primary endpoint of investigator-assessed rPFS, 24.8 months versus 16.6 for the control arm, with a statistically and clinically significant difference. This equated to a 34% reduction in the risk for radiographic progression with the PARPi. Furthermore, the rPFS benefit of add-on olaparib was observed both in patients with and without HRR gene mutations (*n* = 226 and 552, respectively), as detected by circulating tumor DNA testing, with significant HRs in favor of olaparib of 0.50 and 0.76, respectively. In the overall study cohort, the addition of olaparib was also associated with significant improvements in the secondary endpoints of time to first subsequent treatment (HR = 0.74) and time to second progression or death (HR = 0.69). OS data were not mature at the time of analysis, as just 28.6% of the study population had died, but a trend favoring olaparib plus abiraterone was observed (HR = 0.86). Adverse events (AEs) of grade 3 or worse occurred in 47.2% of patients in the olaparib group and 38.4% of those in the placebo group, with anemia being the most common of them (15.1 vs. 3.3%). The rate of discontinuation of olaparib was higher than that of placebo, at 13.8% versus 7.8%, but a comparable proportion of patients in both groups discontinued abiraterone, at 8.5% and 8.8%, respectively. Additionally, the mortality rate due to AEs was also similar, at 4.0% and 4.3% [78].

One of the great differences between these two combination trials is that olaparib’s benefit seems independent of patients’ HRR status, while niraparib’s is restricted to HRR-mutated cancer.

Importantly, these results have some immediate implications: firstly, for the moment, olaparib is propelled as the PARPi of choice for mCRPC; secondly, these results move PARPi to the first line of mCRPC; and, thirdly, the results demonstrated a benefit in patients, irrespective of the homologous recombination repair (HRR) status, thus conferring an advantage over other PARPi.

There is currently a large number of ongoing clinical trials investigating these combinations in other clinical settings than the mCRPC, such as the mHSPC, nonmetastatic castration-resistant disease, and in high-risk, nonmetastatic/localized PCa patients [79], whose results we will be obtaining in the coming years, and which are summarized in Table 2.

#### 3.4.2. Combinations with Immune Checkpoint Inhibitors (ICI)

There are two mechanisms of action that justify the investigation of the combination of PARPi and ICI inhibitors in prostate cancer. The first is the hypothesis that tumors deficient in HRR can result in an increase in neoantigen loads due to the accumulation of indels and frameshift mutations [80,81]. The second is that dsDNA breaks accumulation as a result of PARP inhibition, which would result in the release of dsDNA out of the nucleus into the cytoplasmic space of the cell, and this could lead to an increased activation of the interferon (IFN) genes (STING) signaling pathway [82].

Some trials are studying this combination. A phase I/II trial of olaparib, plus the NTI-PD L1 durvalumab in men with mCRPC after abiraterone or enzalutamide showed a median rPFS of 16.1 months (95% CI 4.5–16.1 months), with most of the responses in patients with BRCA2 mutation [83]. Cohort A of the phase Ib/II KEYNOTE-365 study (NCT02861573) has the purpose of assessing the clinical activity of olaparib combined with the anti-PD-1 pembrolizumab on 104 postdocetaxel mCRPC patients who progressed after at least two lines of androgen receptor signaling inhibitor (ARSi). Enrolled patients did not have detectable HRR gene alteration.

Evan Yu and colleagues have displayed the interim results of cohort A patients with a median follow-up of 19.3 months during the European Society for Medical Oncology (ESMO) Congress 2021. The combination appeared to improve PSA RR regardless of HRR status if compared with olaparib or pembrolizumab monotherapies in the same setting [84].

Recently, the Phase III KEYLYNK-010 trial [85] investigating pembrolizumab in combination with olaparib in patients with mCPRC who progressed after treatment with chemotherapy and either AAP or enzalutamide has been stopped after an interim analysis in which the combination did not demonstrate a benefit in OS, one of the study’s dual primary endpoints, compared to the control arm of either AAP or enzalutamide.

Other ongoing clinical trials are presented in Table 3.

#### 3.4.3. Combination with Anti-VEGF Therapies

The rationale for the combination between PARPi and antiangiogenic drugs is based in the fact that hypoxic conditions have shown to impair synthesis of homologous recombination proteins in vitro [86]. This effect would cause a HRR defect that can be treated with PARPi. A randomized phase II trial combining olaparib plus cediranib in mCRPC showed a rPFS, the primary endpoint, clearly prolonged for patients receiving the combination [87].

#### 3.4.4. Combination with AKT Inhibitors/ATR Inhibitors

Among the many PARPi resistance mechanisms, the PARP-1/Akt interaction in oxidative stress may explain the potential efficacy of PARPi + AKT-inhibitor combinations. Therefore, the inhibition of the Akt pathway might prevent this kind of PARPi resistance [88]. The NCT03840200 is currently investigating the coadministration of rucaparib and the AKT-inhibitor ipatasertib.

The combination of PARP inhibition and ATR inhibition is a new therapeutic strategy, which is today assessed in different forms of cancer. The TRAP trial (NCT03787680) is an ongoing phase II study comparing the responses of mCRPC patients with DDR mutations to those of mCRPC patients without DDR mutations, who are all treated with olaparib plus the ATR-inhibitor AZD6738 [89].

#### 3.4.5. Combination with DNA-Damaging and Other DDR-Targeting Agents

The combination of PARPi with DNA-damaging chemotherapies is challenging due to overlapping hematological toxicities. The combination veliparib–temozolomide is the only one reported, showing modest results [90].

#### 3.4.6. Combination with Radionuclides

It has been suggested in preclinical cancer models that radiometabolic therapies could have a synergistic antitumoral activity if added to ADT and PARPi. Radium-223 is an α-emitting radioisotope that induces DNA double-stranded breaks leading to cell death, and some early-phase clinical trials for molecularly unselected patients with mCRPC tested its combination with PARPi. The safety profile of niraparib and radium-223 was the primary endpoint of the single-arm NiraRad study (NCT03076203), which recruited patients with mCRPC and bone metastases, concluding that the combination is safe and tolerable [91] and the phase I/II COMRADE trial (NCT03317392) is evaluating radium-223 with or without olaparib in patients with bone metastases.

However, in the last few years, a novel class of radiopharmaceuticals (the so-called “rad-hybrid PSMA ligands” or “rhPSMA”) has been developed [92,93].

Due to their specific features, these rhPSMA ligands can be evaluated both for imaging and therapy in PCa, being one of the treatments that are creating more expectations in PCa. PSMA-targeted small molecules bound to 177-Lutetium (177Lu) delivering high doses of beta-radiation after internalizing in the cancer cell, causing double-strand breaks [94,95].

Some trials are studying the combination of 177Lu-PSMA radioligands with PARPi [96]. Ongoing trials with this and other combination of drugs are shown in Table 4.

### 3.5. PARPi and Radiotherapy

Interestingly, when combined with ionizing radiation, PARP inhibition has been found to enhance cellular radiosensitivity [97]. Radiation induces physical and biochemical DNA damage. Cells respond by activating three main repair mechanisms: two double-stranded DNA breaks (DSB) repair pathways ((nonhomologous end-joining, (NHEJ), homologous recombination, (HR)), and a single-stranded DNA breaks (SSB) repair pathway (base excision repair, BER) [98].

In contrast to the more easily repaired SSB, DSB are highly mutagenic and cytotoxic if unrepaired. However, SSB can be converted to DSB in the context of DNA replication if unrepaired, interfering with important cellular processes and survival. Importantly, SSB are more common in the context of external beam radiotherapy. Therefore, emerging evidence indicates that PARPi can act as radiosensitizers through the BER pathway, increasing the risk of collapsed replication forks, and thus generating persistent DSBs [99]. Furthermore, studies have shown that in addition to HR and NHEJ, DSBs may also be repaired by an alternative end-joining pathway (Alt-EJ), which requires PARP1 [100,101,102]. Consequently, PARPi use in cells that have switched to Alt-EJ has demonstrated to enhance radiosensitivity [103].

Several PARPi have been investigated in preclinical studies. Rae et al. studied the effects of olaparib and rucaparib in combination with radiation in prostate cancer cells. They found a similar cytotoxic effect with half of the radiation dose when the cells were exposed to PARPi on both androgen-dependent and androgen-resistant lines [104]. Recent evidence suggests the existence of close crosstalk between DDR machinery and androgen hormone-signaling pathways mainly based on exposure to ionizing radiation.

AR axis is activated by radiation-induced DNA double-strand lesions in PCa malignant cells, thus leading to the upregulation of several DDR genes. The androgen-deprivation strategy may induce the downregulation of these DDR genes, promoting an increased malignant cell death. Therefore, the activity of PARP is increased, and this latter phenomenon leads to the tumor-cell survival and the modulation of AR-axis activity [72]. This is the biological rationale of the combined use of PARPi, ADT, and radiation-based therapies. In this setting, promising phase II trials are ongoing, such as the NADIR study investigating niraparib with standard combination radiation therapy and androgen deprivation therapy in patients with high-risk PCa (NCT04037254) [105].

If these findings are confirmed, they could be of great importance, since it would allow the dose of radiotherapy to be reduced, thereby minimizing its adverse effects.

## 4. Mutation Testing.: Whom, When, and How

Germline and somatic mutations are frequent in localized and advanced PCa. Detection of these mutations may help to identify patients with a more aggressive evolution and to offer them the opportunity for individualized treatment [106].

The IMPACT study was designed to perform PSA screening in carriers of germline BRCA 1/2 mutation. The results revealed that a PSA limit lower than usual to indicate biopsy (3 ng/mL) allowed early detection of prostate cancer in this population [107,108].

The identification of families with this germline mutation would also allow screening of other types of cancer for which they have an increased predisposition [109,110].

Therefore, germline and somatic mutation testing in men with PCa could guide treatment options [111]. Thus, since the approval of PARPi in PCa treatment, PCa guidelines recommend germline testing of mutations in DNA repair genes [112].

The National Comprehensive Cancer Network (NCCN) PCa Early Detection guidelines recommend PCa screening at age 40 years for BRCA2 carriers, and consider for individuals with other germline mutations beginning shared decision-making about PSA screening at age 40 years and screening at annual intervals rather than every other year [113].

The NCCN PCa guidelines recommend germline multigene testing in the following scenarios: metastatic, regional (node positive), very-high and high-risk localized prostate cancer, family history and/or ancestry of different cancers, especially if diagnosed <50 and personal history of breast cancer. Intermediate-risk prostate cancer with intraductal/cribriform histology and a prior personal history of some cancers.

Somatic tumor testing for alterations in homologous recombination DNA repair genes is recommended in metastatic PCa, and in regional PCa. Tumor testing for microsatellite instability-high or deficient mismatch repair in mCPRC and in regional or castration-naive metastatic PCa. TMB testing may be considered in patients with mCRPC [114].

The ESMO clinical practice guidelines for PCa recommend early PSA testing in BRCA1/2 carriers >40 years, and germline testing for BRCA2 and other DDR genes associated with cancer predisposition syndromes in men with a family history of cancer, and in all patients with metastatic prostate cancer. In addition, it’s recommended to consider tumor testing for HR genes and mismatch repair defects (or microsatellite instability) in mCRPC [115]. The European Association of Urology guidelines also recommend germline testing in metastatic PCa, and also consider it in high-risk PCa who have a family member diagnosed with PCa before 60 years, multiple family members diagnosed with PCa at age <60 years or a family member who died from PCa, and in men with a family history of high-risk germline mutations or a family history of multiple cancers on the same side of the family, but the exact extent and sequence of testing remain debatable [3,116].

The most appropriate method for detecting an HRR deficiency has not been determined either, although tumor testing is currently considered the gold standard. In PCa, tumor testing with next-generation sequencing (NGS) is the most widely available method of detecting a germline HRRm or somatic HRRm, as well as copy-number changes and genomic instability [115]. Recently, a tiered approach was proposed for determining the most appropriate sample to identify DDR alterations using NGS [117]. Fresh metastatic tumor biopsy was deemed optimal, followed by ctDNA. If none is available, an archival sample of metastatic or primary tumor tissue may be used, although there are some drawbacks about: disease is often multifocal, and the metastatic clone could be not represented in the core biopsy analyzed, and the degradation of DNA after years in paraffin. Lastly, blood or saliva samples may be considered for germline testing. Although this may miss important somatic alterations in DDR-related genes and cannot discern monoallelic from biallelic aberrations, the reported PARPi response rates for germline and somatic mutations are comparable [118]. Nonetheless, germline-only testing might miss approximately half of BRCA1/2 alterations and patients with microsatellite instability.

## 5. Conclusions and Future Perspectives

PARPi have become the first targeted therapy approved for biomarker-selected patients with advanced PCa. The current approval is in monotherapy for the second-line setting or later. However, the promising perspectives of the ongoing studies have led to the design of several trials looking at new settings and strategies: earlier stages of the disease and combination therapies.

Defects in HRR genes have been identified in 20–25% of patients with advanced prostate cancer, but the therapeutic significance of these defects is not completely understood. Studies with PARPi have demonstrated their antitumor activity but do not clarify the optimal gene set to include as a selection biomarker. Responses are frequent among patients with BRCA1/2 mutations and less common among patients with ATM or CDK12 alterations. It is more difficult to generate robust data on low-prevalence mutations, since it is very difficult for any trials to accumulate sufficient cases.

In the coming years, future research will be aimed at solving some essential questions: identifying the patient profile that will benefit most from this treatment, recognizing the best moment to treat, and optimizing the combination therapies.

Soon, as more results are obtained from the ongoing clinical trials, we will witness the approval for PARPi at different settings of PCa. Hopefully, that would benefit a considerable number of patients.

## Figures and Tables

**Figure 1 biomedicines-10-01416-f001:**
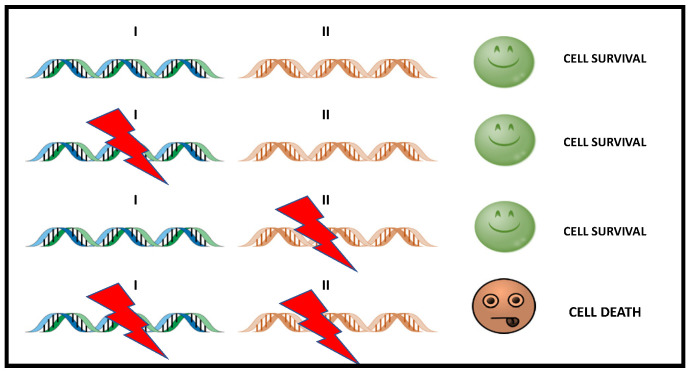
Simple scheme of Synthetic Lethality. The alteration of either gene alone (gen I or gen II) does not causes cell death, while the simultaneous alteration of two genes triggers cell death. In cancer treatment, gene II becomes a therapeutic target that can be used to attack tumor cells with dysfunction in I.

**Figure 2 biomedicines-10-01416-f002:**
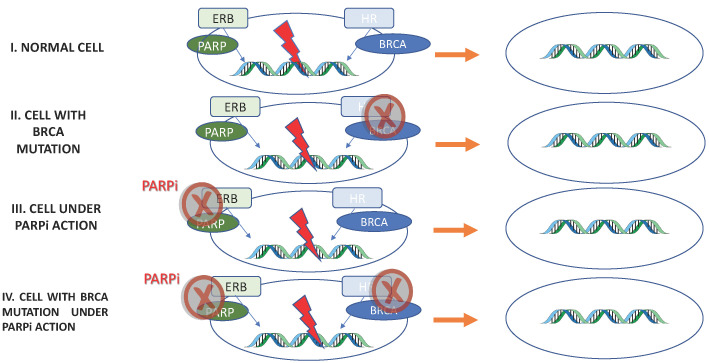
PARPi antitumor activity based on Synthetic Lethality. I: Normal cell. Excision repair base (ERB) and homologous recombination (HR) are functional: repair DNA maintaining cell viability. II, III: Through BRCA mutation or PARPi, one of the repair pathway’s is inhibited. Once the other pathway is functional, the cell maintains its viability. IV: Both DNA repair pathways are inhibited; therefore, errors in the DNA are not repaired, resulting in cell death.

**Figure 3 biomedicines-10-01416-f003:**
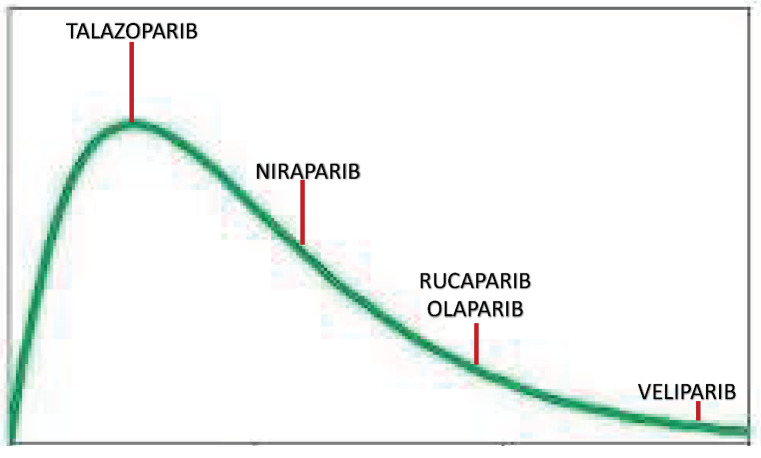
PARPi Trapping Potency.

**Table 1 biomedicines-10-01416-t001:** Chemical structures and pharmacological properties of the currently available PARP inhibitors (PARPi).

	Structural Formula	Dose [mg]	Cmax [ng/mL]	Tmax [h]	Half-Life [h]	PARP1 Trapping Ability	Direct Off-Targets	Indirect Off-Targets
Olaparib (AZD-2281, MK-7339)	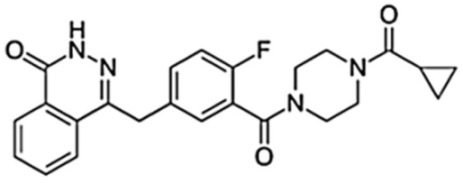	300/12 h	7700	1.5	14.9	Moderate	PARP1, PARP2, PARP3	
Rucaparib (AG014699)	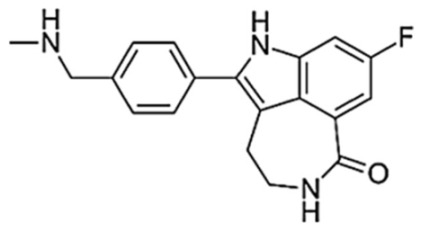	600/12 h	1940	1.9	25.9	Moderate	ARTD5, ARTD6, PARP1,2,3 TNKS1, TNKS2	ALDH2, H6PD, CDK16, PIM3, DYRK1B
Niraparib (MK-4827)	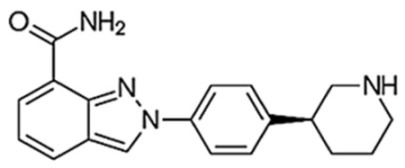	300/24 h	2232	3.0	36	Moderate–high	PARP2, PARP3, PARP4, PARP12	ALDH2, CIT, DCK, DYRK1A, DYRK1B
Talazoparib (BMN-673)	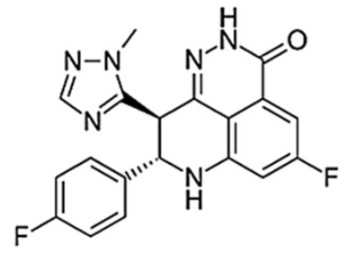	1/24 h	16.4	1–2	90	Very high	PARP2 PARP1	
Veliparib (ABT-888)	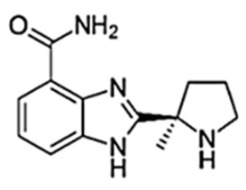	40/12 h	410	1.0	6	Very low	PARP2, PARP3, PARP10	

**Table 2 biomedicines-10-01416-t002:** Phase I, II, or III trials assessing PARP inhibitors in combination with Antiandrogen therapy. Abbreviations: mCRPC = metastatic castration-resistant prostate cancer; mHSPC = metastatic hormone sensitive prostate cancer; enza = Enzalutamide; AAP = abiraterone plus prednisone; RT = radiotherapy; PFS = progression free survival; rPFS = radiological progression free survival; OS = overall survival; DLT = dose-limiting toxicity.

STUDY	DESIGN	Estimated Enrollment	Setting	Agent(s)	Homologous Recombination Repair Mutations	Primary Endpoint(s)
TALAPRO-2 NCT03395197	Phase III, randomized	872	mCRPC treatment naïve	Talazoparib + enza vs. placebo + enza	Selected	Safety, PFS
TALAPRO-3 NCT04821622	Phase III, randomized	550	mHSPC	Talazoparib + enza vs. placebo + enza	Selected	rPFS
NCT03012321	Phase II, randomized	70	mCRPC treatment naïve	Olaparib vs. AAP vs. olaparib + AAP	Selected	PFS
PROpel NCT03732820	Phase III, randomized	720	mCRPC treatment naïve	Olaparib + AAP vs. placebo + AAP	Unselected	rPFS
CASPAR NCT04455750	Phase III, randomized	1002	mCRPC treatment naive	Rucaparib + enza vs. placebo + enza	Unselected	rPFS, OS
MAGNITUDE NCT03748641	Phase III, randomized	1000	mCRPC treatment naïve	Niraparib + AAP vs. placebo + AAP	Selected	rPFS
NADIR NCT04037254	Phase II, randomized	180	High risk localized or locally advanced PCa (no prior treatment)	Niraparib + RT + ADT vs. niraparib alone vs. RT + ADT	Unselected	Maintenance of disease-free state
ASCLEPIuS NCT04194554	Phase I/II, single arm, open label	100	High risk locally advanced PCa (cN+)	Niraparib + AAP + leuprolide + RT	Unselected	DLT, biochemical failure (% of pts)
AMPLITUDE NCT04497844	Phase III, randomized	788	mHSPC	NIRAPARIB+AAP vs. PLACEBO+AAP	Selected	rPFS
ZZ-First (NCT04332744)	Phase II, randomized, open label	54	mHSPC	Enzalutamide vs. enzalutamide + talazoparib	Unselected	PSA-Complete Response

**Table 3 biomedicines-10-01416-t003:** Phase I, II, or III trials assessing PARP inhibitors in combination with immune checkpoint inhibitors (ICI). Abbreviations: mCRPC = metastatic castration-resistant prostate cancer; nmCRPC = nonmetastatic castration-resistant prostate cancer; ARSi = androgen receptor signaling inhibitor; enza = enzalutamide; AAP = abiraterone plus prednisone; ORR = objective response rate; AEs = adverse effects; rPFS = radiological progression free survival; OS = overall survival.

STUDY	DESIGN	Estimated Enrollment	Setting	Agent(s)	Homologous Recombination Repair Mutations	Primary Endpoint(s)
KEYNOTE- 365, NCT02861573	Phase Ib/II, nonrandomized	1000 (104 in COHORT A)	mCRPC after docetaxel and one prior ARSi	Olaparib + pembrolizumab (cohort A)	Unselected	PSA response rate, ORR, safety
KEYLYNK-010, NCT03834519	Phase III, randomized	780	mCRPC after docetaxel and one prior ARSi	Olaparib + pembrolizumab vs. enza/AAP	Unselected	OS, rPFS
NCT03810105	Phase II, single arm	32	Biochemically recurrent nmCRPC	Olaparib + durvalumab	Selected	Number of pts with undetectable PSA
CheckMate 9KD, NCT03338790	Phase II, nonrandomized	330	mCRPC chemotherapynaïve	Nivolumab + rucaparib/ enza/docetaxel	Selected	ORR, PSA response rate
QUEST, NCT03431350	Phase Ib/II, multiarm, nonrandomized	150	mCRPC after prior CT and ARSi (depending on cohorts)	Niraparib + AAP vs. niraparib + JNJ-63723283 (anti-PD1)	Both selected and unselected	ORR, incidence of AEs

**Table 4 biomedicines-10-01416-t004:** Phase I, II, or III trials assessing PARP inhibitors in combination with other molecules and Radionuclides. Abbreviations: mCRPC = metastatic castration-resistant prostate cancer; ARSi = androgen receptor signaling inhibitor; CT = chemotherapy; DLT = dose-limiting toxicity; rPFS = radiological progression free survival; CR = complete response; PR = partial response.

STUDY	DESIGN	Estimated Enrollment	Setting	Agent (s)	Homologous Recombination Repair Mutations	Primary Endpoint(s)
LuPARP, NCT03874884	Phase I, single arm	52	mCRPC after prior CT and ARSis	Olaparib + 177Lu-PSMA	Not available	Dose-limiting toxicity (DLT) recommended phase II dose
COMRADE, NCT03317392	Phase I/II, randomized	112	mCRPC after prior CT and ARSis	Olaparib + Radium-223 vs. Radium-223	Not available	rPFS, maximum tolerated dose
NiraRad, NCT03076203	Phase Ib, single-arm	14	mCRPC after at least one prior ARSi, with or without prior CT	Niraparib + Radium-223	Unselected	DLT
TRAP, NCT03787680	Phase II, nonrandomized	47	mCRCP after prior ARSi	Olaparib + AZD6738 (ATRinhibitor)	Selected	Rate of response (CR or PR), PSA response >50% decline
NCT03840200	Phase Ib, nonrandomized	51	mCRPC after one prior ARSi	Rucaparib + ipatasertib (AKTinhibitor)	Unselected	DLT, PSA response rate
NCT02893917	Phase II, randomized	90	mCRPC after at least one prior therapy	Olaparib + cediranib (VEGF-R TKI) vs. olaparib	Not available	rPFS

## Data Availability

Not applicable.

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
