# Peer review of "PARP Inhibitors: A New Horizon for Patients with Prostate Cancer"

_biomedicines, 2022, doi:10.3390/biomedicines10061416_

Round 1
Reviewer 1 Report
Introduction is too simple, a more complex presentation of prostate cancer pathology, symptomatology and treatment options is needed
Chemical structures and structural characterization of the substances is missing
Comparison between substances would be interesting and also comparison with current therapeutic options
The template seems that it was not used correctly throughout the whole manuscript (please check)
Author Response
Response to Reviewer 1 Comments
We greatly appreciate your comments, and we are sure that they will enrich the final manuscript.
Point 1: Introduction is too simple, a more complex presentation of prostate cancer pathology, symptomatology and treatment options is needed
Response 1: Please provide your response for Point 1. (in red)
As you advise, we have written a new, more complex introduction, providing data on the nature of prostate cancer, the different current treatment options, and those that have been incorporated in recent years, the mechanisms of resistance to hormonal therapies, and therefore justify the introduction of new drugs with different mechanisms of action such as PARPi.
We have also introduced some data on the genetics of prostate cancer and the concept of synthetic lethality, which will have so much relevance later in the main text. Likewise, we have also introduced a figure where the concept of synthetic lethality is represented in a simple way (because it is the introduction).
We enclose the new introduction, with the changes in red color.
INTRODUCTION
Prostate cancer (PCa) is the most common tumor diagnosed in men in the Western World, the third cause of cancer-related death among men in Europe, and the second in the United States [1,2].
Fortunately, most prostate cancers are diagnosed as localized diseases and can be treated successfully with radical prostatectomy or radiotherapy. However, more than 5% of patients are diagnosed with metastatic disease, and 30-40% of patients will develop biochemical recurrence after treatment intended to cure. A considerable proportion of them will develop metastatic castration-resistant prostate cancer (mCRPC), a condition that leads to death in a short period of time [3].
PCa can be defined as an heterogenous condition, which ranges from a relatively indolent to an aggressive disease.
Primary PCa is characterized by heterogeneity. The disease is often multifocal, and distinct foci can arise as clonally separate lesions with little or no shared driver gene alterations. In contrast, the metastatic foci of PCa seem to arise from a single clone and manifest subclonal homogeneity [4].
Androgen receptor (AR) signaling plays a central role in the disease’s development; therefore, chemical castration, or androgen deprivation therapy (ADT), remains the mainstay of systemic treatment [5].
Several mechanisms try to explain resistance to this therapy, such as androgen receptor (AR) gain-offunction mutations or splice variants (e.g., AR-V7), loss of tumor suppressor genes (e.g., p53, pTEN), and modifications of stromal components into the tumor microenvironment, promoting invasion, neoangiogenesis, and metastatisation process [6].
In recent decades, a large number of novel drugs with different mechanisms of action have been introduced in the therapeutic management against prostate cancer in all stages, focusing mainly on metastatic and castration-resistant disease, with and without metastasis, where have shown to delay progression and prolong survival. Clinical options include cytostatic agents (docetaxel plus prednisone, cabacitaxel), second-generation antiandrogens, (abiraterone, enzalutamide, apalutamide and darolutamide) bone-targeted therapies (Radium 223), immunotherapy (sipuleucel-T), Akt inhibitors, and radioisotopes (Lutetium-177–PSMA-617). However, there is a lack of high-quality rabdomized trials exploring different treatment sequences.
Therefore, there is an urgent need for more effective treatments for patients with mCRPC, especially those who have progressed after novel hormonal therapies and taxane chemotherapies.
Recently, two poly (ADP-ribose) polymerase (PARP) inhibitors (Olaparib and Rucaparib) have been approved for mCPRC, opening a new horizon in the treatment of PCa [7].
Olaparib was approved for mCRPC patients with any Homologous Recombination (HR) mutation, who had previously received a second-generation hormonal agent, and rucaparib was approved for mCRPC patients with a BRCA1 or BRCA2 mutation, who had previously received both a second-generation hormonal agent and a taxane chemotherapy.
The rationale for the use of PARP inhibitors (PARPi) is mainly based on the high frequency of genetic mutations found in PCa and the interesting phenomenon of synthetic lethality.
Genetic alterations in PCa include germline and somatic mutations. Germline mutations affect all cells of the body and may be useful for genetic counseling. Somatic alterations occur only in tumor cells and may evolve due to intrinsic genome instability within the tumor and clonal selection resulting from earlier therapies.
Germline mutations in Homologous Recombination (HR) DNA repair genes have been observed in approximately 10–15% of patients with metastatic PCa, while somatic mutations occur in 20–25% of those patients, being BRCA2 and ATM the most frequently mutated.
Moreover, compared to locoregional tumors, mCRPC presents a higher frequency of aberrations in AR, TP53, RB1, and PTEN genes. [8]. These evidences support the research of PARPi in CPRCm. Nevertheless, these genetic alterations have also been described in localized disease, conferring a more aggressive evolution. Therefore, research with PARPi is being extended to earlier stages of the disease.
Synthetic lethality is a genetic concept in which the functional loss of two genes results in cell death, while the functional loss of only one of them is compatible with cell viability (Figure 1).
BRCA1 and BRCA2 are tumor suppressor genes involved in transcriptional regulation and repair of double-strand DNA damage breaks (DSBs) in the DNA molecule, playing a role key on the HR path. Cells with loss of function in these genes are unable to repair errors in DNA, depending on the ability of PARP to activate alternative routes. Since they depend on PARPs to maintain genome integrity, these cells are highly vulnerable to PARPi.
There are still many unanswered questions regarding the use of PARPi in PCa, including who would benefit more from a genetic study and when is the best time to use it in the natural evolution of the disease [9]. These questions make it necessary to deepen the knowledge of these drugs to find potential ways to maximize their clinical effectiveness.
This review aims to update the available evidence on PARPi for PCa treatment.
FIGURE 1. Simple scheme of Synthetic Lethality. (See figure 1 in the attachment)
The alteration of either gene alone (gen I or gen II) does not causes cell death, while the simultaneous alteration of two genes triggers cell death. In cancer treatment, gene II becomes a therapeutic target that can be used to attack tumor cells with dysfunction in I.
Point 2: Chemical structures and structural characterization of the substances is missing
Response 2: We include a new table (Please see table 1 in the attachment), that represents the chemical structures and pharmacological properties of the currently available PARP inhibitors
Point 3: Comparison between substances would be interesting and also comparison with current therapeutic options
Response 3: In the previous table that we have incorporated (table 1) , main differences between different PARPi are specified, and it is also emphasized in the section where the mechanism of action of the different PARPi is explained, specifically one of the most relevant characteristics, their ability to trap PARP1;
We have also added a new figure, figure 3, that represents the different ability of each PARPi to trap PARP1 on DNA.
(please see PDF Figure 3 in the attachment)
A comparison between different PARPi is carried out in the section on toxicity, where the differences between PARPi are highlighted.
However, no direct comparisons between different PARPi or other current therapeutic options have been done within one clinical trial.
Point 4: The template seems that it was not used correctly throughout the whole manuscript (please check)
Response 4: We have not used the template because it was not mandatory, but a recommendation. However, if using it makes it easier to review and you prefer, we include the new manuscript in Biomedicines Microsoft Word template with changes in red colour.
Please see the attached document

Reviewer 2 Report
This review needs major revision before its consideration to publication.
- Introduction of the manuscript is too concise and short summary of concept of synthetic lethality should also be included in introduction portion to make a connecting link between PARPi and synthetic lethality concept.
- The manuscript is mare a collection of data rather generating any information or informative discussion of the data. The reviewer suggests to reconsider whole manuscript and try to make it more informative and insightful.
- There are no schematic illustrations in the manuscript, the reviewer suggests to at least incorporate figures which depicts the whole conclusion and rationale of the manuscript.
- The abbreviations used in this manuscript are random. Please use single abbreviation for a particular word throughout the manuscript.
- The referencing style of whole manuscript is not properly managed. The reviewer suggests to use single referencing style and insert reference at specific positions
Author Response
We greatly appreciate your comments, and we are sure that they will enrich the final manuscript.
We also include the new manuscript in Biomedicines Microsoft Word template (with changes in red colour) as a reviewer 1 recommendation.
Point 1: Introduction of the manuscript is too concise and short summary of concept of synthetic lethality should also be included in introduction portion to make a connecting link between PARPi and synthetic lethality concept.
Response 1:
As you advise, we have written a new, more complex introduction, providing data on the nature of prostate cancer, the different current treatment options, and those that have been incorporated in recent years, mechanisms of resistance to hormonal therapies, and therefore justify the introduction of new drugs with different mechanisms of action such as PARPi.
We have also introduced some data on the genetics of prostate cancer and the concept of synthetic lethality, which will have so much relevance later in the main text. Likewise, we have also introduced a figure where the concept of synthetic lethality is represented in a simple way (because it is the introduction).
We enclose the new introduction with the changes in red for easy identification.
INTRODUCTION
Prostate cancer (PCa) is the most common tumor diagnosed in men in the Western World, the third cause of cancer-related death among men in Europe, and the second in the United States [1,2].
Fortunately, most prostate cancers are diagnosed as localized diseases and can be treated successfully with radical prostatectomy or radiotherapy. However, more than 5% of patients are diagnosed with metastatic disease, and 30-40% of patients will develop biochemical recurrence after treatment intended to cure. A considerable proportion of them will develop metastatic castration-resistant prostate cancer (mCRPC), a condition that leads to death in a short period of time [3].
PCa can be defined as an heterogenous condition, which ranges from a relatively indolent to an aggressive disease.
Primary PCa is characterized by heterogeneity. The disease is often multifocal, and distinct foci can arise as clonally separate lesions with little or no shared driver gene alterations. In contrast, the metastatic foci of PCa seem to arise from a single clone and manifest subclonal homogeneity [4].
Androgen receptor (AR) signaling plays a central role in the disease’s development; therefore, chemical castration, or androgen deprivation therapy (ADT), remains the mainstay of systemic treatment [5].
Several mechanisms try to explain resistance to this therapy, such as androgen receptor (AR) gain-offunction mutations or splice variants (e.g., AR-V7), loss of tumor suppressor genes (e.g., p53, pTEN), and modifications of stromal components into the tumor microenvironment, promoting invasion, neoangiogenesis, and metastatisation process [6].
In recent decades, a large number of novel drugs with different mechanisms of action have been introduced in the therapeutic management against prostate cancer in all stages, focusing mainly on metastatic and castration-resistant disease, with and without metastasis, where have shown to delay progression and prolong survival. Clinical options include cytostatic agents (docetaxel plus prednisone, cabacitaxel), second-generation antiandrogens, (abiraterone, enzalutamide, apalutamide and darolutamide) bone-targeted therapies (Radium 223), immunotherapy (sipuleucel-T), Akt inhibitors, and radioisotopes (Lutetium-177–PSMA-617). However, there is a lack of high-quality rabdomized trials exploring different treatment sequences.
Therefore, there is an urgent need for more effective treatments for patients with mCRPC, especially those who have progressed after novel hormonal therapies and taxane chemotherapies.
Recently, two poly (ADP-ribose) polymerase (PARP) inhibitors (Olaparib and Rucaparib) have been approved for mCPRC, opening a new horizon in the treatment of PCa [7].
Olaparib was approved for mCRPC patients with any Homologous Recombination (HR) mutation, who had previously received a second-generation hormonal agent, and rucaparib was approved for mCRPC patients with a BRCA1 or BRCA2 mutation, who had previously received both a second-generation hormonal agent and a taxane chemotherapy.
The rationale for the use of PARP inhibitors (PARPi) is mainly based on the high frequency of genetic mutations found in PCa and the interesting phenomenon of synthetic lethality.
Genetic alterations in PCa include germline and somatic mutations. Germline mutations affect all cells of the body and may be useful for genetic counseling. Somatic alterations occur only in tumor cells and may evolve due to intrinsic genome instability within the tumor and clonal selection resulting from earlier therapies.
Germline mutations in Homologous Recombination (HR) DNA repair genes have been observed in approximately 10–15% of patients with metastatic PCa, while somatic mutations occur in 20–25% of those patients, being BRCA2 and ATM the most frequently mutated.
Moreover, compared to locoregional tumors, mCRPC presents a higher frequency of aberrations in AR, TP53, RB1, and PTEN genes. [8]. These evidences support the research of PARPi in CPRCm. Nevertheless, these genetic alterations have also been described in localized disease, conferring a more aggressive evolution. Therefore, research with PARPi is being extended to earlier stages of the disease.
Synthetic lethality is a genetic concept in which the functional loss of two genes results in cell death, while the functional loss of only one of them is compatible with cell viability (Figure 1).
BRCA1 and BRCA2 are tumor suppressor genes involved in transcriptional regulation and repair of double-strand DNA damage breaks (DSBs) in the DNA molecule, playing a role key on the HR path. Cells with loss of function in these genes are unable to repair errors in DNA, depending on the ability of PARP to activate alternative routes. Since they depend on PARPs to maintain genome integrity, these cells are highly vulnerable to PARPi.
There are still many unanswered questions regarding the use of PARPi in PCa, including who would benefit more from a genetic study and when is the best time to use it in the natural evolution of the disease [9]. These questions make it necessary to deepen the knowledge of these drugs to find potential ways to maximize their clinical effectiveness.
This review aims to update the available evidence on PARPi for PCa treatment.
FIGURE 1. Simple scheme of Synthetic Lethality.
The alteration of either gene alone (gen I or gen II) does not causes cell death, while the simultaneous alteration of two genes triggers cell death. In cancer treatment, gene II becomes a therapeutic target that can be used to attack tumor cells with dysfunction in I.
(See Figure 1 in the attached text)
Point 2: The manuscript is mare a collection of data rather generating any information or informative discussion of the data. The reviewer suggests to reconsider whole manuscript and try to make it more informative and insightful.
Response 2: Following your advice, we have incorporated some of our own opinions throughout the text to make it more informative and useful.
We have incorporated an informative discussion of the data in the following sections:
- INTRODUCTION
- CLINICAL DEVELOPMENT OF PARP INHIBITORS IN PROSTATE CANCER
- ADVERSE EVENTS AND TOLERABILITY
- PARPi IN COMBINATION
- MUTATION TESTING. WHOM, WHEN, AND HOW
- CONCLUSIONS AND FUTURE PERSPECTIVES
that we hope will be of your interest.
(Please see attached text)
Point 3: There are no schematic illustrations in the manuscript, the reviewer suggests to at least incorporate figures which depicts the whole conclusion and rationale of the manuscript.
Response 3: As you recommend, we incorporate figures 1, 2 and 3.
Figure 1, in the INTRODUCTION, is a simple scheme of the concept of Synthetic Lethality (Please see figure 1 in the attached text)
Figure 2, in (MECHANISM OF ACTION OF PARP INHIBITORS), to explain PARPi antitumor activity based on Synthetic Lethality.
(Please see figure 2 in the attached text)
Figure 3, in (MECHANISM OF ACTION OF PARP INHIBITORS), to explain different PARPi TRAPPING POTENCY
(Please see figure 3 in the attached text)
Furthermore, as a recommendation from reviewer 1, we have also added a new table, TABLE 1, (Please see table 1 in attached text), that represents the chemical structures and pharmacological properties of the currently available PARP inhibitors
Point 4: The abbreviations used in this manuscript are random. Please use single abbreviation for a particular word throughout the manuscript.
Response 4: We have corrected and supervised all the abbreviations so they are all homogeneous throughout the manuscript and coincide with the abbreviations published in the literature, and have defined the first time they appear in each of the three sections: abstract; the main text; the first figure or table (Please check changes in the manuscript). If you detect any particular fault, please let us know.
Point 5: The referencing style of whole manuscript is not properly managed. The reviewer suggests to use single referencing style and insert reference at specific positions
Response 5: We have corrected and supervised all the references conform to the magazine´s recommendation (Journal Articles: 1. Author 1, A.B.; Author 2, C.D. Title of the article. Abbreviated Journal Name Year, Volume, page range.), (Please check changes in the manuscript). If you detect any particular fault, please let us know.

Round 2
Reviewer 1 Report
The quality of the article improved, the authors commplied with reviewer's recommandation
The article can be published in the current form
Reviewer 2 Report
Authors have responded to the comments. The manuscript is acceptable in present form.